# How Microbiota-Derived Metabolites Link the Gut to the Brain during Neuroinflammation

**DOI:** 10.3390/ijms231710128

**Published:** 2022-09-04

**Authors:** Jessica Rebeaud, Benjamin Peter, Caroline Pot

**Affiliations:** Laboratories of Neuroimmunology, Service of Neurology and Neuroscience Research Center, Department of Clinical Neurosciences, Lausanne University Hospital, University of Lausanne, 1066 Lausanne, Switzerland

**Keywords:** neuroinflammation, multiple sclerosis, experimental autoimmune encephalomyelitis, gut–brain axis, microbiota-derived metabolites

## Abstract

Microbiota-derived metabolites are important molecules connecting the gut to the brain. Over the last decade, several studies have highlighted the importance of gut-derived metabolites in the development of multiple sclerosis (MS). Indeed, microbiota-derived metabolites modulate the immune system and affect demyelination. Here, we discuss the current knowledge about microbiota-derived metabolites implications in MS and in different mouse models of neuroinflammation. We focus on the main families of microbial metabolites that play a role during neuroinflammation. A better understanding of the role of those metabolites may lead to new therapeutical avenues to treat neuroinflammatory diseases targeting the gut–brain axis.

## 1. Introduction

Multiple sclerosis (MS) is the most common debilitating neurological disease affecting the central nervous system (CNS) in young adults [1]. More than 2.5 million people worldwide suffer from MS with a higher prevalence in women than men [2]. MS is characterized by chronic demyelination of the CNS by infiltrating self-reactive myelin-specific T cells [3] and by an increased permeability of the blood–brain barrier (BBB). The main symptoms observed in persons with MS (pwMS) are fatigue, vision loss, numbness, cognitive defects, depression, and bladder/bowel dysfunctions [4]. Three main types of MS exist: relapsing–remitting (RR) MS, secondary progressive (SP) MS, and primary progressive (PP) MS. A total of 85% of pwMS suffer from RRMS, which is characterized by episodes of neurological symptoms (relapses) followed by a recovery phase (remissions). Ten to twenty years after RRMS diagnosis, almost 80% of patients progress to the SPMS form of the disease [2]. SPMS is defined by the initial RRMS course followed by a progression of neurological symptoms, sometimes accompanied by relapses. PPMS strikes a smaller percentage of pwMS, and is characterized by a gradual worsening of neurological functions in the absence of relapse [5]. MS lesions in the CNS can be active or inactive. Active lesions are characterized by ongoing inflammation, whereas inactive lesions consist of demyelination and neuronal loss caused by an earlier immune attack.

Animal models contribute to the unravelling of the mechanisms underlying MS pathogenesis and therapy development. Experimental autoimmune encephalomyelitis (EAE) is a widely used animal model to study neuroinflammation. The most common EAE model is a model of relapsing–remitting MS and uses C57BL6 mice [6]. In MS and its animal models, immune cells, both from the adaptive and innate immunity, contribute to inflammation and its resolution [7]. We focus mainly on the adaptive immunity implicated in the gut–brain axis, in particular on CD4^+^ T lymphocytes. Those lymphocytes can be subdivided based on their cytokine profiles in both pro- and anti-inflammatory subsets. Since the original classification by Mosmann and Coffman of CD4^+^ helper T (Th) lymphocytes into Th1 and Th2 subsets [8], the repertoire of CD4^+^ T cell subsets has expanded. Th1 cells are essential for eliminating intracellular pathogens, and Th17 cells induce immunity against extracellular bacteria and fungi. Interestingly, Th17 cells are found in high numbers in the intestinal lamina propria [9]. Furthermore, an exaggerated Th17 response promotes autoimmunity, and elevated levels of IL-17 are detected in MS. Moreover, CD4^+^ T regulatory T cell (Tregs) subsets are important players in inflammation resolution and suppress the activation of effector T cells.

The etiology of MS is not fully unraveled, but it is now widely accepted that its origin is multifactorial. Several contributing environmental factors include smoking, past viral infections (in particular EBV infection), exposure to organic solvents, and low vitamin D intake. Over the last decade, changes in gut environment and microbiota have been pointed out as emerging environmental factors involved in MS development [10,11,12]. Indeed, several studies demonstrated that gut microbiota and the gut–brain axis impact neuroinflammation in MS and its animal model [13,14,15,16]. The gut and the brain are interconnected by several means such as the vagus nerve, the immune cells trafficking between the gut and the brain, the microbiota, and microbiota-derived metabolites. Here, we focus on the last two aspects.

The human gastrointestinal tract represents the largest interface between host, antigens, and environmental factors [17]. Gut microbiota encompasses all microorganisms present in the gastrointestinal tract: bacteria, viruses, archaea, and fungi. Gut microbiota is mainly inherited from the mother during delivery and through breastfeeding. Diet and contact with the external environment continue to shape gut microbiota during the first years of life [18]. In adolescence, it reaches its maximal diversity and remains stable throughout adult life [4]. Gut microbiota has several pivotal roles; it shapes the mucosal immune system, maintains gut barrier integrity, modulates gut neuromuscular functions, and carries out many metabolic functions [18]. The highest concentration of immune cells is found within the gut mucosa. Moreover, gut mucosa, immune cells, and microbiome communicate with each other and play an important role in shaping immune responses throughout the body [19]. Additionally, the gut compartment is a possible location for the generation, expansion, and activation of effector T cells implicated in brain autoimmunity [20,21]. Intestinal dysbiosis is observed in EAE as well as in pwMS [22]. *Bacteroidetes* [10], *Clostridia* clusters XIVa and IV [10], *Faecalibacterium prausnitzii* [23], and *Parabacteroidetes distasonis* [14] levels are significantly decreased in pwMS compared to controls. Conversely, other bacteria, such as *Akkermansia* spp. and *Acinetobacter*, are increased [13,14]. Interestingly, *Akkermansia muciniphila* and *Acinetobacter calcoaceticus* induce pro-inflammatory responses in both human peripheral blood mononuclear cells and monocolonized mice [14]. Conversely, *Parabacteroides distasonis* promotes regulatory T cell (Treg) expansion [14]. Finally, two separate studies show that components of the human gut microbiota participate in autoimmunity of the CNS. Indeed, fecal microbiota transplantation (FMT) from MS patients worsens the course of EAE and reduces the proportion of Treg in mice [13,14]. Gut microbiota components, or their metabolites, can act upon the pro- versus anti-inflammatory T cells, in particular the Th17/Treg ratio, which plays a key role in MS pathogenesis. Accordingly, the intestinal environment may enhance Th17 cell pathogenicity and capacity to trigger brain autoimmunity [24]. In this review, we focus on the main microbiota-derived metabolites known to affect MS and EAE development.

## 2. Lipid Metabolism

### 2.1. Short-Chain Fatty Acids

Fatty acids (FAs) are classified according to the length of their carbon chain. Short-chain fatty acids (SCFAs) are metabolites produced by the fermentation of non-digestible dietary fibers by bacteria in the distal gut lumen whereas medium-chain and long-chain fatty acids (M–LCFAs) originate directly from the diet and are not metabolized by the gut microbiota. Thus, we focus on SCFAs, which are small molecules and are classified according to their carbon number: formate (C1), acetate (C2), propionate (C3), butyrate (C4), and valerate (C5). Acetate, propionate, and butyrate are the most common SCFAs in the human gut as they encompass more than 95% of the total SCFAs [25]. Levels of SCFAs are dependent on the number of indigestible fibers contained in the diet [25]. Indeed, a high-fiber diet containing more non-digestible dietary fibers will lead to elevated SCFA levels. Moreover, the Western diet contains only approximately 10% of indigestible dietary fibers [26], corresponding to the production of 400–600 mmol of SCFAs/day [27].

Germ-free mice have a lower intestinal content of SCFAs than control mice, highlighting the importance of gut bacteria in the formation of SCFAs [28]. Miller and Wolin first highlighted bacterial pathways responsible for SCFA production in the human colon [29]. Several different bacterial species promote SCFA production. Furthermore, acetate, butyrate, and propionate fermentation rely on distinct gut bacterial species [30]. Most of the enteric bacteria, such as *Akkermansia muciniphila, Bacteroidetes* spp., or *Prevetolla* spp., are acetate producers [30,31]. Propionate is produced mainly by bacteria from *Bacteroidetes* and *Firmicutes* phyla [31]. Bacteria from the *Firmicutes* phylum are also implicated in butyrate synthesis [31].

SCFAs can act as signaling molecules. Propionate, butyrate, and valerate are capable of histone deacetylase (HDAC) inhibition [32,33]. Inhibition of HDAC by SCFAs increases acetylation of genes and influences gene transcription. SCFAs are also G-protein-coupled receptor (GPCR) ligands [34]. SCFAs can cross the blood–brain barrier (BBB) and modulate brain function [35]. SCFAs, particularly butyrate, are an important source of fuel for healthy colonocytes [36]. Furthermore, they maintain gut barrier integrity by increasing the expression of tight junction proteins, as well as expanding the proportion of intestinal Treg cells [37].

Reduced fecal and circulating levels of acetate, propionate, and butyrate are observed in pwMS compared to healthy controls (Figure 1) [38,39,40,41,42]. Particularly, Becker et al. found reduced SCFA fecal levels in women affected by MS [43]. Additionally, in a prospective study comparing pregnant women with MS and healthy subjects, a higher propionate/acetate ratio during the first trimester of pregnancy was associated with higher inflammatory activity and an increased relapse rate [44]. Thus, the propionate/acetate ratio could be used as a biomarker for disease activity in pregnant women with MS.

Furthermore, bacterial species and genes involved in butyrate production are decreased in pwMS [45]. PwMS also have lower levels of SCFA-producing bacteria than healthy controls [10,11,12,23,46]. Additionally, propionate supplementation in a small cohort of pwMS restored the Th17/Treg ratio, highlighting the anti-inflammatory role of SCFAs. Additionally, long-term propionate supplementation led to a reduced annual relapse rate and slowed disease progression [39].

Several studies using the mouse model of EAE confirmed the potential beneficial effect of SCFAs on inflammation [33,47,48,49,50,51,52]. Mechanistically, SCFAs have several immunomodulatory properties on both adaptive and innate immune responses, as well as on the non-hematopoietic compartment, which is directly in the central nervous system, specifically. Indeed, propionate supplementation ameliorates the EAE disease course by inducing the differentiation of Tregs and IL-10 production in the spleen and the spinal cord [47]. Furthermore, acetate, propionate, and butyrate supplementation also ameliorate EAE disease severity [48,49,50,51]. Reduced disease severity is linked with an expansion of IL-10 producing Tregs and a reduction in Th1 and Th17 cells. Interestingly, valerate inhibits Th17 cells and IL17A secretion in vitro and upregulates IL10 [33], whereas M- and LCFAs enhance Th1 and Th17 cells differentiation in vitro (46). Furthermore, treatment with valerate reduces the severity of EAE [33]. Propionate supplementation is sufficient to rescue a LCFA diet-induced worsening of EAE disease severity by increasing the number of Tregs and reducing Th17 cell levels [52]. SCFAs further affect B lymphocyte function. Butyrate and propionate interfere with antibody production by B lymphocytes [53], and valerate induces IL-10 production from Bregs [33]. SCFAs affect innate immune cells, in particular neutrophils, by inhibiting pro-inflammatory factor secretion [54], in addition to macrophages by promoting their anti-inflammatory properties [55]. In addition, SCFAs can affect non-hematopoietic compartments, such as the blood–brain barrier (BBB), that are composed of endothelial cells, astrocytes, and pericytes. Butyrate improves BBB permeability in germ-free mice [56]. Butyrate can also affect myelination. Indeed, in the cuprizone-induced demyelination model, a non-inflammatory model of EAE, butyrate ameliorates remyelination and suppresses demyelination [57]. While most SCFA effects are beneficial and dampen neuroinflammation, a dual effect of SCFAs has been reported [48], suggesting that SCFAs could, in addition, promote the generation of inflammatory T cells [38]. Further studies are thus needed to fully elucidate the role of SCFAs. Altogether, SCFAs display several anti-inflammatory and neuroprotective properties that are beneficial in targeting neuroinflammatory diseases.

### 2.2. Bile Acids

Bile acids are liver-produced cholesterol-derived steroids. Bile acids are essential in supporting digestion by acting as detergents in the intestine and absorbing fatty acids, nutrients, and vitamins [58]. Bile acids further act as signaling molecules through their interaction with farnesoid X receptor (FXR) and G-protein-coupled bile acid receptor 1 (GPBAR1).

Primary bile acids are synthesized in the liver and are conjugated either to glycine or taurine to increase their solubility. Conjugated primary bile acids are stored in the gallbladder and are released into the duodenum and support digestion by solubilizing dietary lipids along the intestine. A total of 95% of the conjugated primary bile acids are reabsorbed and recycled in the liver through enterohepatic circulation [58]. Colonic gut bacteria transform the remaining 5% bile acids into secondary bile acids [59]. Two successive steps are needed for secondary bile acids biotransformation: deconjugation and dehydroxylation. Deconjugation is mediated by bacteria which have a bile salt hydrolase activity: this includes bacteria from all major bacterial phyla [60]. Fewer bacterial species of the Firmicutes phylum as *Clostridium* (Cluster XIVa, XI) and *Eubacterium* can dehydroxylate unconjugated bile acids to form secondary bile acids [61]. Germ-free mice display an accumulation of primary bile acids in the gallbladder and reduced levels of secondary bile acids in the colon, highlighting the role of gut bacteria in the metabolism of secondary bile acids [62].

Both bile acid receptors FXR and GBPAR1 are expressed in MS active lesions: FXR is expressed on macrophages, whereas GPBAR1 is expressed on astrocytes and macrophages [63]. FXR knock-out mice display a more severe EAE than wild-type controls [64]. Additionally, treatment of mice with FXR agonists, obeticholic acid [64], or GW4064 [65] reduces EAE disease severity; however, treatment with an abundant primary bile acid such as the chenodeoxycholic acid (CDCA), which is a natural FXR ligand, does not [64]. Interestingly, administration of FXR agonist GW4064 at the beginning of neurological symptoms is sufficient to dampen EAE disease severity (Figure 1) [65]. Reduced EAE disease severity is mediated by elevated IL-10 secretion and the induction of anti-inflammatory macrophages [65]. In addition, FXR expression is downregulated in peripheral immune cells of pwMS compared to healthy controls [65]. However, in vitro FXR-activated human monocytes from pwMS and healthy controls show an increased IL-10 secretion, highlighting the potential therapeutic effect of FXR agonists as an MS treatment [65]. In the same line as FXR agonists, GBPAR1 agonist treatment also dampens EAE disease severity [66]. However, GBPAR1 agonist ameliorates EAE disease by a different mechanism: it reduces immune cell infiltration in the CNS, as well as monocytes’ pro-inflammatory profile and microglial activation [66]. Taken together, both bile acid receptors are interesting targets for the dampening of neuroinflammation.

Furthermore, bile acid metabolism is altered in both pediatric and adult pwMS compared to healthy controls [63], as well as in EAE [67]. Interestingly, bacteria of the genus *Clostridium* (Cluster XIVa and XI), which are reduced in pwMS [10], generate secondary bile acids [61]. The reduction in such bacteria in pwMS could be a potential explanation for the altered bile acid metabolism observed in pwMS. Bile acids can have a direct impact on resident cells of the CNS. On one hand, primary bile acids can have deleterious effects. Indeed, increased concentrations of the primary bile acid taurochenodeoxycholic acid (TCDCA) can disturb BBB integrity in rats [68]. On the other hand, secondary bile acids exert anti-inflammatory actions. TUCDA or UCDA treatment in vitro blocks the neurotoxic polarization of astrocytes, and inhibits the pro-inflammatory polarization of microglia [63,69,70]. Furthermore, TUCDA supplementation in mice ameliorates the EAE disease course (Figure 1) [63]. Demyelination, which is a hallmark of MS, can potentially lead to increased levels of bile acid precursors in the cerebrospinal fluid of pwMS [71]. Nevertheless, higher levels of circulating bile acids are associated with less deterioration in clinical disability in a metabolomic study among pwMS [72]. With this in mind, oral administration of TUCDA has now been tested in a clinical trial involving progressive MS patients with low bile acid levels (Phase 1 and 2 trial (NCT03423121)). Altogether, targeting bile acid metabolism via the gut metabolism opens new avenues of treatment for pwMS.

## 3. Amino acid Metabolism

### 3.1. Tryptophan Metabolism

L-Tryptophan (Trp) is an essential aromatic amino acid that is acquired solely from dietary sources. Trp is required for normal growth and protein synthesis, and serves as a precursor for many bioactive compounds, such as serotonin, niacin, kynurenine, and indole derivatives. Only a small fraction of Trp available is used for protein synthesis. Indeed, the majority of ingested Trp is metabolized through the kynurenine pathway [73]. Trp 2–3 dioxygenase (TDO), indoleamine 2–3 dioxygenase (IDO) 1, and IDO2 are enzymes responsible for the metabolism of Trp into kynurenine. TDO is exclusively present in the liver, while IDO1 is present in the gut and other organs. Under normal conditions, TDO mediates Trp transformation into kynurenine. However, IDO1 can be induced by IFN-γ [74] and modulated by the gut microbiota [75]. Trp metabolism has immunomodulatory functions via the kynurenine pathways. Indeed, Trp is metabolized by several enzymes into kynurenine and other aryl hydrocarbon receptor (AHR) ligands. AHR is a transcription factor and is translocated to the nucleus upon activation, with various ligands obtained from the diet, the environment, and endogenous origins. AHR signaling is implicated in the generation of Treg [76] and type-1 regulatory T cells [77]. Trp can also be metabolized into serotonin and melatonin. Finally, bacteria of the gut microbiota can directly metabolize Trp into indole and its derivatives. Many bacterial metabolites of Trp are AHR ligands, such as tryptamine or indole derivatives, including indole-3-lactate, indole-3-acrylate, indole-3-acetate, indole-3-propionate, indole-3-aldehyde, indoxyl-3-sulfate, or skatole. Some indole derivatives were completely absent from the circulation of germ-free mice, highlighting the importance of the gut microbiota in Trp metabolism [78]. Many bacteria are involved in Trp metabolism: for example, bacteria possessing a tryptophanase activity can convert Trp into indole [79]. Indole can be further metabolized into indole-acidic derivatives or into indole-3-aldehyde by *Lactobacillus reuteri* and *Lactobacillus johnsonnii* [80].

Bacteria involved in the metabolism of indole-3-lactate are reduced in the gut microbiota of pwMS (Figure 1) [45]. Furthermore, indole-3-lactate and indole-3-propionate are significantly lower in the serum of pwMS [45]. Indole-3-lactate can be considered a precursor of indole-3-propionate as indole-3-lactate can be further metabolized into the potent neuroprotective metabolite indole-3-propionate [81]. Additionally, higher serum levels of Trp and higher relative abundance of indole-3-lactate are associated with a lower risk of developing pediatric MS [82]. Conversely, in another study, relapsing MS patients had higher indole-3-propionate levels in their urine than pwMS showing no relapse signs [83].

Furthermore, pwMS have significantly lower urine concentrations of kynurenine and a lower kynurenine/Trp ratio than healthy controls [83]. The kynurenine/Trp ratio is negatively correlated with a higher Expanded Disability Status Scale (EDSS), a disability scale for pwMS [83]. On the contrary, the kynurenine/Trp ratio is increased in the serum of pwMS compared to healthy controls [84]. Kynurenate, which has neuroprotective action, is also increased in relapse–remitting pwMS compared to controls [84]. Conversely, decreased levels of kynurenate in the CSF of pwMS are observed [85]. Interestingly, the quinolinate/kynurenate ratio correlates strongly with EDSS severity and is elevated in PPMS and SPMS [84]. Another study showed that pwMS display an elevated quinolinate/kynurenate ratio in both CSF and the blood [86]. An elevated quinolinate/kynurenate ratio favors neurotoxicity. Indeed, quinolinate mediates excitotoxicity at the NMDA receptor and kynurenate is neuroprotective by antagonizing quinolinate excitotoxicity [87,88]. Interestingly, Trp and metabolites of the kynurenine pathway (kynurenate, quinolinate, picolinate) are potential blood biomarkers that could discriminate MS subtypes [84]. Additionally, CSF metabolites could differentiate relapse–remitting MS patients from secondary progressive pwMS [89]. Particularly, Trp and phenylalanine metabolisms are altered in SP pwMS compared to RRMS [89]. Herman et al. also found a strong association of the bacterial Trp metabolite indole-3-acetate with disease duration [89]. Furthermore, indoxyl sulfate levels positively correlate with neurofilaments (NFL) levels, a marker of disease activity in the CSF of pwMS [85]. NFL measurement in CSF and in the blood is a novel biomarker to assess neurodegeneration. Indoxyl sulfate is neurotoxic and causes axonal damage and neuronal dysfunction in vitro [85].

Trp and kynurenine metabolisms can have an important impact on the EAE disease course. Indeed, Platten et al. first showed that a synthetic Trp metabolite can suppress EAE disease [90]. Since then, several Trp metabolites were shown to have a beneficial impact on the EAE disease course. Indeed, treatment with AHR agonist 3-hydroxyanthranilate (3-HAA) is sufficient to inhibit T cell response, enhance Treg frequency, and dampen EAE disease severity [91]. Furthermore, systemic administration of cinnabarinate, another kynurenine metabolite, suppresses EAE disease by reducing Th17 cells and increasing Treg cells [92]. Treatment with two neuroprotective Trp metabolites, N-acetylserotonin and melatonin, reduce EAE disease severity [93]. Finally, tryptamine administration also dampens EAE disease severity [94].

Oral supplementation with Trp suppresses Th1-specific response during EAE but has no impact on neuroinflammation [95]. Conversely, abrogation of Trp from the diet completely abolishes EAE disease in mice raised in a conventional animal facility (with intact gut microbiota), but not in germ-free mice highlighting the importance of gut microbiota in Trp metabolism [96]. Trp-free diet induces profound changes in the gut microbiota reducing particularly bacteria from *Akkermansia*, *Lactobacillus*, and *Barnesiella* genus [96].

Interestingly, AHR ligands levels in the circulation are significantly lower in pwMS than in controls [97]. Furthermore, AHR ligands are increased in patients with active MS lesions compared to patients with inactive MS lesions, indicating that anti-inflammatory AHR ligands may be produced upon inflammation [97]. Levels of AHR-ligands negatively correlate with MS disease severity [97]. Finally, in mice, AHR activation by endogenous Trp ligands promotes the differentiation of Treg cells, induces tolerogenic dendritic cells, and attenuates EAE disease [98]. On the contrary, AHR activation by indoxyl-3-sulfate treatment stimulates Th17 differentiation, and worsens the EAE disease course [99].

EAE disease severity is increased in mice deficient in IDO1 [91,100]. However, pharmacological inhibition of IDO1 ameliorates EAE disease severity [101]. The role of IDO1 in MS and EAE is, however, not clearly established. IDO1 is the rate-limiting enzyme that controls the metabolism of Trp into kynurenine and its metabolites. Enhanced activity of IDO1 can lead to elevated concentrations of kynurenine metabolites that are both neuroprotective and neurotoxic. On one hand, higher levels of quinolinate could be deleterious for EAE or MS development and IDO1 inhibition could lower quinolinate production and, therefore may dampen EAE disease severity. On the other hand, IDO1 deficiency could reduce levels of neuroprotective Trp metabolites such as kynurenate and, therefore, worsen the EAE disease course.

### 3.2. Phenylalanine and Tyrosine Metabolism

Phenylalanine (Phe) is an essential aromatic amino acid. Tyrosine (Tyr) is synthetized from hydroxylating Phe. Tyrosine hydroxylase is the rate-limiting enzyme that metabolizes Tyr into catecholamines such as dopamine, epinephrine, and norepinephrine.

Both Phe and Tyr are metabolized through fermentation by bacteria in the gut into phenolic compounds [102]. P-cresol sulfate and p-cresol are two Phe phenolic derivatives and are absent in germ-free mice, showing the importance of the gut microbiota in the production of both compounds [78]. Many bacterial species are involved in the formation of p-cresol from tyrosine, such as the *Clostridium difficile*, *Clostridium scatalogenes*, *Proteus vulgaris,* and *Lactobacillaceae* species [103,104]. Interestingly, p-cresol sulfate is elevated in the serum of pwMS [45]. However, levels of bacteria involved in p-cresol synthesis are similar between pwMS and controls [45]. Furthermore, the relative abundance of p-cresol sulfate and phenylacetylglutamate, two bacterial metabolites derived from Phe and Tyr metabolism, is higher in pwMS compared to controls [85]. Both metabolites correlate with NFL levels in the CSF of MS patients and display a neurotoxic activity in vitro [85]. In vitro treatment of splenocytes with p-cresol and p-cresol sulfate reduces IFNγ secretion and increases IL-4 secretion [105]. Furthermore, induced overproduction of p-cresol in mice leads to a reduced Th1 response and an increased Th2 response [105].

Phe metabolism is altered in SPMS and RRMS compared to controls (Figure 1) [89]. Interestingly, Phe is slightly decreased in pwMS displaying active lesions compared to pwMS with no disease activity [106]. Furthermore, Phe concentration is decreased in the CSF of pwMS compared to healthy controls [107]. Phe and Tyr metabolisms are disturbed in EAE urine samples compared to controls [108].

## 4. Trimethylamine N-Oxide (TMAO)

Gut microbiota bacteria from phyla *Actinobacteria*, *Bacteroidetes*, *Firmicutes*, and *Proteobacteria* [109] metabolize choline, carnitine, betaine, and ergothioneine from the diet into trimethylamine (TMA). TMA is then transported from the gut to the liver where it is oxidized into TMAO, which is further excreted in the urine [110].

High plasma TMAO levels promote thrombosis in cardiovascular patients [111]. However, lower serum levels of the precursor of TMAO carnitine were observed in MS patients compared to healthy controls [45]. Furthermore, dietary supplementation with L-carnitine is beneficial in the mouse model of amyotrophic lateral sclerosis (ALS), a degenerative neurological disease [112]. Conversely, carnitine and betaine concentrations were higher in ALS patients compared to healthy controls, while TMAO and choline were lower [113]. The impact of TMAO and its precursors on MS development and its EAE animal model should be investigated as it could be an interesting target. Furthermore, the underlying mechanisms remains largely unraveled.

## 5. Polyphenols Metabolism

Dietary polyphenols are antioxidant metabolites present in a wide range of food such as wine, tea, coffee, beer, fruits, vegetables, extra virgin olive oil, and chocolate [114]. Many dietary polyphenols exhibit antioxidative as well as anti-inflammatory activities. Dietary polyphenols metabolized by the gut microbiota are urolithins A and B, phenylacetate, phenylpropionate, valerate, valerolactone, and phloroglucinol [115]. *Clostridium* and *Eubacterium* genera are involved in the biotransformation of dietary polyphenols [115].

Urolithins are gut microbiota metabolites of ellagitannins, which are bioactive polyphenols. Ellagic acid and ellagitannins-rich foods are, for example, berries, pomegranate, walnuts, and almonds [116]. *Gordonibacter* and *Ellagibacter* contribute to urolithins’ production in the colon [117,118]. Urolithin A oral administration alleviates active and passive EAE disease severity in mice and reduced both demyelination and CNS cell infiltration [119]. Furthermore, urolithin A in vitro treatment inhibits Th17 polarization by targeting AHR [119]. Supplementation with ellagic acid dampens EAE disease severity in rats [120] and in mice [121]. Mice orally fed with pomegranate peel extract display less severe EAE symptoms than the control mice [122]. Indeed, pomegranate peel extract supplementation suppresses CNS inflammation and infiltration of immune cells [122]. Treating mice with a new formulation of pomegranate peel extract also reduces EAE disease severity [123]. However, immune cell infiltration and inflammation in the spinal cord are comparable between treated and untreated mice [123]. Interestingly, fecal microbiota transplantation from EAE mice, supplemented with pomegranate peel extract, to naïve EAE mice is sufficient to delay EAE onset for 2 days [122]. Furthermore, pomegranate peel extract treatment modifies gut microbiota composition and significantly increases the relative abundance of *Lactobacillaceae* while reducing *Alcaligenaceae* and *Acidaminococcaceae* [122]. In vitro treatment with urolithin A extracted from whole raspberries reduces inducible nitric oxide synthase (iNOS) expression and polarizes microglia into an M2-phenotype [124]. Furthermore, in the same study, urolithin A from raspberries dampens LPS-induced neuroinflammation [124].

Interestingly, a synthetic derivative of the microbiota-derived metabolite phloroglucinol is neuroprotective. Indeed, EAE disease severity is dampened in rats upon treatment with the phloroglucinol derivative [125]. Additionally, Th1 and Th17 cell infiltration in the CNS is reduced upon treatment [125]. Furthermore, in the cuprizone-induced demyelination model, the phloroglucinol derivative significantly improves remyelination [125].

Other dietary polyphenols that are not directly metabolized by the gut microbiota can modulate gut microbiota composition and have an impact on neuroinflammation. Curcumin [126], hesperidin [127], resveratrol [128], apigenin [129], and daidzein [130] supplementation reduce EAE disease severity. Taken together, dietary polyphenols have a beneficial impact on neuroinflammation. Furthermore, an isoflavone-rich diet (isoflavone dietary polyphenols are daidzein, genistein, and equol [114]) ameliorates EAE disease severity in mice [131].

## 6. Polyamines

Polyamines are small molecules, having two or more amino groups that are derived from L-arginine metabolism. Polyamines derive from various sources, such as alimentation, production by resident gut bacteria, or endogenous synthesis [132]. The most common polyamines comprise spermine, spermidine, and putrescine. Many bacteria can synthesize polyamines [133] that play an important role both on the immune and the nervous systems [134]. Furthermore, polyamine metabolism is involved in T cell lineage commitment [135] and is associated with Th17 cell pathogenicity [136]. Taking these results together, polyamines could be interesting targets in MS pathology.

Spermidine is present in the CSF of both pwMS and healthy controls [137]. PwMS display elevated polyamine synthesis compared to controls [138]. Indeed, the blood of pwMS exhibits higher levels of enzymes involved in polyamines synthesis: arginine decarboxylase, ornithine decarboxylase, and agmatinase [138].

Spermidine treatment alleviates EAE disease severity in mice by significantly reducing demyelination in the CNS [139,140]. Furthermore, reduced lymphocyte infiltration in the CNS of spermidine-treated mice is observed [140]. Interestingly, the reduction in EAE disease severity is mediated by the induction of inhibitory macrophages by spermidine [140]. Additionally, in vitro treatment of Th17 with spermidine shifts their phenotype towards regulatory FoxP3^+^ T cells, both in murine and human cell culture [141]. Spermine treatment also dampens EAE disease severity and reduces CNS infiltration of immune cells, as well as demyelination [142]. Interestingly, spermine treatment specifically reduces the population of pathogenic CD4^+^ T cells expressing IFNγ and/or IL-17 in vivo during EAE [142]. Furthermore, in vitro treatment of CD4^+^ T cells blocks their activation and proliferation through inhibition of the MAPK/ERK pathway by spermine [142].

## 7. Bacterial Peptidoglycan

The above-described metabolites are broken down by the gut microbiota. We will now discuss other gut-derived metabolites independent of this mechanism. First, metabolites are derived directly from bacterial structures. Peptidoglycans (PGNs) are components (sugars and amino acids) of the bacterial cell wall that are recognized by pattern-recognition receptors (PRRs) as PGRPs and PGLYRP1-4 [143]. PGNs are common to almost all bacteria and are not restricted to pathogens. Growing evidence highlights the importance of PGNs in maintaining host homeostasis and promoting developmental processes [144]. PGNs signal through Toll-like receptors (TLRs) and NOD-like receptors. PGNs are present in both macrophages and dendritic cells in the white matter of pwMS in higher numbers compared to controls [145]. PwMS display elevated levels of antibodies against PGNs in CSF, suggesting an intrathecal production of antibodies in MS [145]. Interestingly, the fractions of microbial reads are greater in the post-mortem brain sections of pwMS than in controls [146]. PGN immunodetection in the white matter of pwMS correlates with demyelination and inflammation [147].

EAE monkeys show higher brain levels of phagocytes (mostly macrophages) that display intracellular PGNs [148]. Interestingly, these cells produce the pro-inflammatory cytokine IL-12 [148]. Furthermore, in vitro treatment of peripheral blood mononuclear cells (PBMCs) from humans and monkeys with purified PGNs extracted from *S.aureus* induce IL-12 secretion [148]. Purified PGNs extracted from *S.aureus* are a strong adjuvant when emulsified in IFAs, and can substitute CFAs in active EAE induction [149]. EAE progression is dependent on dendritic cell activation by PGNs present in the CNS through the NOD-1, NOD-2, and RIP-2 mediated pathways [150]. Finally, neutralization of circulating PGNs ameliorate EAE disease severity in mice [151]. Taken together, these results highlight the potential of modulating PGN levels in circulation to dampen neuroinflammation.

## 8. Non-Ribosomal Peptides and Polyketides

In addition, essential but less characterized gut microbiota-derived metabolites can be directly synthesized by the gut microbiota, including non-ribosomal peptides (NRP) and polyketides (PK). These metabolites are viewed with increased significance but remain understudied in the context of EAE and MS. NRP and PK are directly synthesized from the gut microbiota through their non-ribosomal peptide synthase and/or their polyketide synthase activities. Interestingly, several NRP and PK derived from plants or microorganisms are already used as therapeutics: NRP are used as antibiotics or immunosuppressants, whereas PK are used as antibiotics in addition to anticancer, anthelmintic, cholesterol-lowering, and antifungal drugs. NRP and PK derived from the mammalian microbiota are less known. However, the PK–NRP hybrid colibactin synthesized from bacteria from *Enterobacteriaceae* is a genotoxin associated with colorectal cancer [152]. Furthermore, the NRP tillivaline is linked with antibiotic-associated hemorrhagic colitis and possesses a cytotoxic activity [153]. Finally, dipeptide aldehyde is a protease inhibitor produced by bacteria from *Clostridium* sp. [154]. Nevertheless, metabolites naturally produced by the mammalian gut microbiota are less characterized than microbiota-derived metabolites and are more difficult to identify and characterize [155]. However, their better characterization is crucial and could be of great importance in finding new therapeutics for MS and other diseases [156].

## 9. Conclusions

In recent decades, several studies have highlighted the importance of microbiota-derived metabolites in fine-tuning immune responses and in the development of MS (Figure 1). Several metabolites can have either beneficial or detrimental effects on disease development. Many gut-derived metabolites and their producers remain unidentified, and further studies should be carried on to better characterize them and decipher their role during neuroinflammation. However, microbiota-derived metabolites are already interesting targets as they can be modulated by diet as well as disease status. Dietary supplementation with beneficial bacterial metabolites or bacterial substrates should be considered in the treatment of MS. Further studies on MS should be conducted to study their potential therapeutical functions.

## Figures and Tables

**Figure 1 ijms-23-10128-f001:**
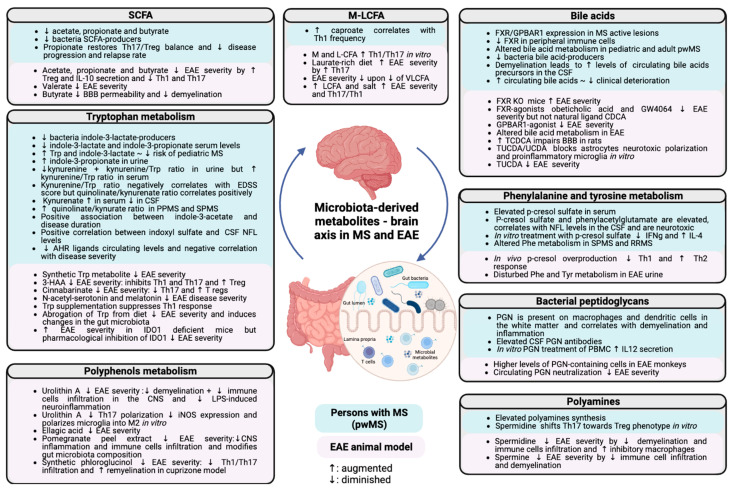
Implications of the different microbiota-derived metabolites in MS pathogenesis. The main concepts described in this review are summarized and separated between human and mice data. SCFAs, short-chain fatty acids; EAE, experimental autoimmune encephalomyelitis; BBB, blood–brain barrier; IL, interleukin; Th, T helper; Treg, regulatory T cell; MCFAs, medium-chain fatty acids; LCFAs, long-chain fatty acids; VLCFAs, very-long-chain fatty acids; FXR, farnesoid X receptor; GPBAR, G-protein-coupled bile acid receptor; CSF, cerebrospinal fluid; KO, knock-out; TCDCA, taurochenodeoxycholic acid; TUCDA, tauroursochenodeoxycholic acid; UCDA, ursochenodeoxycholic acid; NFL, neurofilaments; IFN, interferon; Phe, phenylalanine; SPMS, secondary progressive multiple sclerosis; RRMS, primary progressive multiple sclerosis; Tyr, tyrosine; PGN, peptidoglycan; PBMC, peripheral blood mononuclear cells; iNOS, inducible nitric oxide; IDO, indoleamine 2-3 dioxygenase; Trp, tryptophan; 3-HAA, agonist 3-hydroxyanthranilate; AHR, aryl hydrocarbon receptor. Augmented: **↑**; diminished: ↓.

## Data Availability

Not applicable.

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
