# Peer review of "How Microbiota-Derived Metabolites Link the Gut to the Brain during Neuroinflammation"

_ijms, 2022, doi:10.3390/ijms231710128_

Round 1
Reviewer 1 Report
The review covered different gut microbiota-derived metabolites and their roles in MS. Overall, the review is scientifically sound, up to date, and well written. However, some irrelevant information is included and can distract the readers. The authors also left out essential microbiota-derived metabolites classes.
Major comments:
- In lines 152-165, MCFA and LCFA are not microbiome-derived and fall out of the scope of the review. I suggest removing section 2.2 entirely.
- In lines 167-189, the authors included excessive information on bile acids that is not closely related to gut microbiota or MS. The paragraphs must be edited and shortened to focus on the review topic.
- The authors covered metabolites that were broken down by the gut microbiota. However, the discussion has left out essential metabolites synthesized by the gut microbiota, namely through nonribosomal peptide synthetases (NRPS) and polyketide synthases (PKS). The discussion, therefore, is not complete and needs to be extended.
Minor comments:
1. In line 420, the authors classified peptidoglycans as cell wall proteins. This description is incorrect. Peptidoglycans consist of sugars and amino acids and are not considered proteins.
2. In line 425, the phrase 'PGN-containing cells' is vague and needs clarification.
3. In Figure 1, I suggest changing 'EAE mouse model' to 'EAE animal model' in the pink box since some sections also contain results from the monkeys.
Author Response
Reviewer 1:
The review covered different gut microbiota-derived metabolites and their roles in MS. Overall, the review is scientifically sound, up to date, and well written. However, some irrelevant information is included and can distract the readers. The authors also left out essential microbiota-derived metabolites classes.
Major comments:
In lines 152-165, MCFA and LCFA are not microbiome-derived and fall out of the scope of the review. I suggest removing section 2.2 entirely.
Response: We have now removed the section 2.2 and only mention the MCFA and LCFA as not microbiome-derived.
In lines 167-189, the authors included excessive information on bile acids that is not closely related to gut microbiota or MS. The paragraphs must be edited and shortened to focus on the review topic.
Response: We have edited and shortened the paragraph to focus on secondary bile acids related to the topic.
The authors covered metabolites that were broken down by the gut microbiota. However, the discussion has left out essential metabolites synthesized by the gut microbiota, namely through nonribosomal peptide synthetases (NRPS) and polyketide synthases (PKS). The discussion, therefore, is not complete and needs to be extended.
Response: We thank the reviewer for this important remark. We have now added a new section 8 dedicated to the discussion of nonribosomal peptide synthetases (NRPS) and polyketide synthases (PKS) that opens new perspectives in the use of metabolites during neuroinflammatory disease that are not yet studied.
Minor comments:
In line 420, the authors classified peptidoglycans as cell wall proteins. This description is incorrect. Peptidoglycans consist of sugars and amino acids and are not considered proteins.
Response: We thank the reviewer for this comment; we have now corrected this mistake (line 417-418).
In line 425, the phrase 'PGN-containing cells' is vague and needs clarification.
Response: the phrase 'PGN-containing cells has been removed and reformulated (lines 429-430).
In Figure 1, I suggest changing 'EAE mouse model' to 'EAE animal model' in the pink box since some sections also contain results from the monkeys.
Response: This has been changed (line 968)

Reviewer 2 Report
Jessica Rebeaud et al. summarized the microbiota-derived metabolites implications in MS and in different mouse models of neuroinflammation, which could provide a better understanding of the role of those metabolites in neuroinflammatory diseases treating. However, there are several concerns that need to be improved.
1. Line 40-41: “The most common EAE model models relapsing-remitting MS using C57BL6 mice [6].” This sentence is difficult to understand, please rewrite it.
2. Line 58-60: “Indeed, many studies demonstrated that gut microbiota and gut-brain axis influence MS course” It is better for the author to cite some literature on the subject rather than a review.
3. Line 116: “SCFAs” changed as “SCFA”.
4. Line 133-134: Please add some references for this sentence “Several studies using the mouse model of EAE confirmed the potential beneficial effect of SCFA in inflammation.”
5. Line 248-250: “Several bacterial phyla are involved in Trp metabolism. For example, bacteria possessing a tryptophanase activity can convert Trp into indole [78]. Indole can be further metabolized into indole-acidic derivatives or into indole-3-aldehyde by Lactobacillus reuteri and Lactobacillus johnsonnii”. maybe bacterial genera??
6. Please briefly summarize the possible mechanism of SCFA, MCFA, LCFA and bile acid on MS in the appropriate place.
7. Similarly, a brief summary of the possible mechanism of Amino acid metabolism, Trimethylamine N-oxide, Polyphenols metabolism, Polyamines, bacterial peptidoglycan on MS is also required.
Author Response
Reviewer 2:
Jessica Rebeaud et al. summarized the microbiota-derived metabolites implications in MS and in different mouse models of neuroinflammation, which could provide a better understanding of the role of those metabolites in neuroinflammatory diseases treating. However, there are several concerns that need to be improved.
Line 40-41: “The most common EAE model models relapsing-remitting MS using C57BL6 mice [6].” This sentence is difficult to understand, please rewrite it.
Response: The sentence has been reformulated (line: 40-41).
Line 58-60: “Indeed, many studies demonstrated that gut microbiota and gut-brain axis influence MS course” It is better for the author to cite some literature on the subject rather than a review.
Response: We now cite the original papers (lines 57-58).
Line 116: “SCFAs” changed as “SCFA”.
Response: This has been changed
Line 133-134: Please add some references for this sentence “Several studies using the mouse model of EAE confirmed the potential beneficial effect of SCFA in inflammation.”
Response: we have now added appropriate references (lines 136-137).
Line 248-250: “Several bacterial phyla are involved in Trp metabolism. For example, bacteria possessing a tryptophanase activity can convert Trp into indole [78]. Indole can be further metabolized into indole-acidic derivatives or into indole-3-aldehyde by Lactobacillus reuteri and Lactobacillus johnsonnii”. maybe bacterial genera??
Response: We have now rephrased this sentence (lines 240-243).
Please briefly summarize the possible mechanism of SCFA, MCFA, LCFA and bile acid on MS in the appropriate place.
Response: we have now summarized possible mechanisms of the different metabolites in MS.
SCFA, MCFA, LCFA: lines 143-161.
Bile acids: lines 203-217
Similarly, a brief summary of the possible mechanism of Amino acid metabolism, Trimethylamine N-oxide, Polyphenols metabolism, Polyamines, bacterial peptidoglycan on MS is also required.
Response: similarly, we have now discussed more specifically the mechanism of action of the different metabolites in each section.
Amino acid metabolism: lines 278-307
Trimethylamine N-oxide: lines 346-347
Polyphenols metabolism: lines 361-363
Polyamines: lines 405-413
The revised manuscript is uploaded with this answer:

Reviewer 3 Report
Comments
This review article entitled "Role of gut microbiota-derived metabolites on neuroinflammation and multiple sclerosis" tried to summarize the gut-derived metabolites in the development of multiple sclerosis (MS). Overall, the paper has clear ideas and abundant references. However, the following issues need to be further addressed to build better quality of reading and comprehension. The specific comments are as follows:
1. Introduction section line 57-58 “Over the last decade, changes in the gut environment and microbiota have been pointed out as emerging environmental factors involved in MS development”. Could the authors please show the exact provenances of this sentence?
2. In 2.2 bile acid section, what is the relationship between aryl hydrocarbon receptor (AHR) and bile acid? Clarification of the relationship between them obviously is very important for subsequent logical reasoning.
Author Response
Reviewer 3:
This review article entitled "Role of gut microbiota-derived metabolites on neuroinflammation and multiple sclerosis" tried to summarize the gut-derived metabolites in the development of multiple sclerosis (MS). Overall, the paper has clear ideas and abundant references. However, the following issues need to be further addressed to build better quality of reading and comprehension. The specific comments are as follows:
Introduction section line 57-58 “Over the last decade, changes in the gut environment and microbiota have been pointed out as emerging environmental factors involved in MS development”. Could the authors please show the exact provenances of this sentence?
Response: We have now reformulated this sentence and added the corresponding references (lines 58-60).
In 2.2 bile acid section, what is the relationship between aryl hydrocarbon receptor (AHR) and bile acid? Clarification of the relationship between them obviously is very important for subsequent logical reasoning.
Response: We thank the reviewer for this important remark. Indeed the link between AHR and bile acid is indirect. It has been proposed that AHR activation by its agonist TCDD alters bile acid metabolism in mice (Csanaky IL et al., Toxicol Appl Pharmacol 2018 Mar 15;343:48-612. 018). However, as this is only an indirect link, we removed this part of the manuscript and focused on AHR on the following section on amino acid metabolism.

Round 2
Reviewer 1 Report
The authors have addressed my concerns and improved the manuscript.